# Study on the Optimization of an Extraction Process of Two Triterpenoid Saponins in the Root of *Rosa laevigata* Michx. and Their Protective Effect on Acute Lung Injury

**DOI:** 10.3390/ph18020253

**Published:** 2025-02-13

**Authors:** Jingya Mo, Qiaoyu Deng, Yuanyuan Huang, Xuegong Jia, Fengfeng Xie, Bei Zhou, Hongwei Gao, Yanchun Wu, Jingquan Yuan

**Affiliations:** 1Zhuangyao Medical Research Institute of Traditional Chinese Medicine, Guangxi University of Traditional Chinese Medicine, Nanning 530200, China; mojingya08@163.com (J.M.); dqylky@163.com (Q.D.); yuanyhuang@yeah.net (Y.H.); jxg101313@163.com (X.J.); zoey619@163.com (B.Z.); 2Guangxi Key Laboratory of Zhuangyao Medicine, Guangxi University of Traditional Chinese Medicine, Nanning 530200, China; 15177143553@163.com

**Keywords:** kajiichigoside F1, rosamultin, process optimization, acute lung injury, mechanism of action, root of *Rosa laevigata* Michx

## Abstract

**Objectives:** Kajiichigoside F1 and rosamultin are natural triterpenoid saponins found in the root of *Rosa laevigata* Michx. These compounds are isomers, making their separation challenging. Nonetheless, they have been reported to exhibit significant anti-inflammatory activity, although their mechanism of action remains unclear. This study aimed to optimize the extraction process of echinacoside and rosamultin from *R. laevigata* and to elucidate the anti-inflammatory mechanisms of these saponins in an LPS-induced acute lung injury (ALI) model. **Methods:** The extraction process was optimized using a single-factor experiment and the Box–Behnken response surface methodology, with the content of kajiichigoside F1, rosamultin, and their total content serving as evaluation indices. The acute lung injury model was induced by LPS, and lung tissue damage was assessed through hematoxylin and eosin (HE) staining. The secretion of relevant inflammatory factors was quantified using enzyme-linked immunosorbent assay (ELISA), and the expression levels of associated proteins were analyzed via Western blotting. **Results:** The optimal extraction conditions were determined to be an ethanol volume fraction of 80.0%, a solid–liquid ratio of 1:25, an extraction duration of 80 min, and three extraction cycles. Kajiichigoside F1 and rosamultin were found to mitigate alveolar inflammation in mice with acute lung injury (ALI) by effectively reducing the expression of the pro-inflammatory cytokines TNF-α and IL-6. Additionally, these compounds down-regulated the expression of phosphorylated NF-κB p65 and NF-κB IκBα proteins, thereby alleviating inflammatory symptoms. **Conclusions:** Kajiichigoside F1 and rosamultin attenuate the inflammatory response in acute lung injury induced by lipopolysaccharide (LPS) stimulation through modulation of the NF-κB signaling pathway. This study preliminarily elucidates their anti-inflammatory mechanism, suggesting that both compounds possess therapeutic potential for ALI. These findings provide significant guidance for the future development of active components derived from the root of *R. laevigata* and establish a foundation for enhancing the quality standards of its medicinal materials.

## 1. Introduction

Acute lung injury (ALI) is a respiratory condition precipitated by various pathogenic factors that result in damage to lung tissue. This disease is characterized by a rapid onset and a high propensity for deterioration, potentially leading to respiratory failure in severe cases [1,2,3]. In recent years, the incidence of ALI has markedly increased due to the sudden outbreak of novel coronavirus pneumonia, with mortality rates reaching 30% to 40%. This rise not only poses a significant threat to human health and well-being but also places an additional strain on the global public health system [4,5,6]. The pathogenesis of ALI is primarily associated with pulmonary edema, inflammatory infiltration, and damage to pulmonary microvascular endothelial cells, all of which are mediated by inflammatory agents [7]. Therefore, a crucial therapeutic approach for ALI involves the effective modulation of the inflammatory response.

The root of *Rose laevigata* Michx., a significant component of Chinese herbal medicine, contributes substantially to the industrial output value of traditional Chinese medicine. It has been traditionally recognized for its therapeutic effects, including the consolidation of essence, intestinal astringency, wind dispelling, dampness removal, detoxification, and detumescence [8]. Contemporary pharmacological research has demonstrated that the root of *R. laevigata* exhibits a range of biological activities, such as enhancing immune function and providing antioxidant, anti-inflammatory, and analgesic effects, as well as possessing anti-tumor, antibacterial, antiviral, hypoglycemic, hypolipidemic, and renal protective properties [9,10,11,12,13,14,15].

The previous investigation conducted by the research group identified a high concentration of kajiichigoside F1 (2*α*, 3*α*, 19*α*-trihydroxyurs-12-en-28-oic-28-O-*β*-D-glucopyranose) and rosamultin (2*α*, 3*β*, 19*α*-trihydroxyurs-12-en-28-oic-28-O-*β*-D-glucopyranose) in the root of *R. laevigata* (The molecular structure is depicted in Figure 1). These compounds are ursane-type pentacyclic triterpenoid saponin differential isomers. The existing literature indicates that they possess various bioactivities, including anti-inflammatory properties, the ability to counteract the toxicity of oncology drugs, and antinociceptive effects [16,17]. Consequently, the Box–Behnken response surface methodology was employed for the first time to optimize the extraction process and determine the optimal conditions for their extraction. Subsequently, the simultaneous preparation of these two saponins was achieved for the first time using D101 macroporous resin, silica gel, C_18_, and other chromatographic techniques. This process yielded 1.1321 g of kajiichigoside F1 and 1.2755 g of rosamultin, each with a purity exceeding 99.0%. The present study examined the mechanisms of action of kajiichigoside F1 and rosamultin in the context of lipopolysaccharide (LPS)-induced acute lung injury (ALI) in BALB/c mice through in vivo experimentation. Histological analysis via hematoxylin and eosin (HE) staining, coupled with enzyme-linked immunosorbent assay (ELISA) and Western blot analysis, demonstrated that both compounds exerted differential ameliorative effects on the pulmonary tissues of mice afflicted with ALI. Notably, treatment with kajiichigoside F1 and rosamultin resulted in a significant reduction in the expression levels of tumor necrosis factor-alpha (TNF-α) and interleukin-6 (IL-6), as well as in the phosphorylation levels of nuclear factor kappa-light-chain-enhancer of activated B cells (NF-κB) p65 and inhibitor of kappa B alpha (IκBα). These results underscore the protective effects of kajiichigoside F1 and rosamultin against LPS-induced acute lung injury and elucidate their underlying mechanisms of action, thereby providing a foundational basis for their further development.

## 2. Results

### 2.1. Experimental Results from Single-Factor Optimization of the Extraction Process

An investigation into three extraction methods—ultrasonic extraction, ethanol reflux extraction, and water decoction—revealed that the yields of kajiichigoside F1 and rosamultin from the root of *R. laevigata* were highest with ultrasonic extraction, as illustrated in Figure 2. Consequently, ultrasonic extraction was selected for further optimization. Subsequent examination of various extraction parameters, including the ethanol volume fraction, solid–liquid ratio, extraction duration, and number of extraction cycles, indicated that the optimal conditions were as follows: utilizing 25 times the volume of 80% ethanol and conducting ultrasonic extraction three times, with each cycle lasting 100 min, as shown in Figure 3, Figure 4, Figure 5 and Figure 6.

### 2.2. The Box–Behnken Response Surface Methodology Was Employed to Optimize the Extraction Process

The Box–Behnken response surface methodology was employed to optimize the extraction process of the root of *R. laevigata*. Design-Expert 8.0.6 software was utilized to fit a multiple regression model to the experimental data. A second-order polynomial regression model was developed to describe the total extraction rate of kajiichigoside F1 and rosamultinin as a function of ethanol volume fraction (A), solid–liquid ratio (B), extraction time (C), and number of extraction cycles (D). The model is expressed as follows: extraction rate (Y = 7.72 + 0.74A + 0.24B − 0.010C + 0.76D + 0.17AB − 0.20AC + 0.079AD − 0.082BC − 0.091BD + 0.11CD − 0.23A^2^ − 0.093B^2^ + 0.10C^2^ − 0.44D^2^). The coefficient of determination (R^2^) is 0.8470, with an F-value of 5.54 and a *p*-value of 0.0014, indicating that the regression model is statistically significant. The lack-of-fit F-value is 1.14 with a *p*-value of 0.4887, suggesting that the lack-of-fit is not significant and that the model is suitable for analyzing the experimental data. The results are presented in Table 1. When considering the response variable Y, the order of influence of the first-order terms in the model is D > A > B > C, indicating that the number of extraction cycles has the greatest effect, followed by ethanol volume fraction, solid–liquid ratio, and extraction time. The first-order items A and D exhibited a highly significant effect (*p* < 0.001), while the second-order item D2 demonstrated a significant effect (*p* < 0.05). However, the interaction item did not have a significant impact on the total content, as presented in Table 2. In conclusion, the response surface analysis of the interactions among various factors was conducted using Design-Expert 8.0.6 software. This analysis facilitated the generation of response surface plots and contour maps, as depicted in Figure 7, which effectively illustrate the interactions among the experimental factors. A steeper response surface indicates a stronger interaction between two factors, thereby exerting a more significant influence on the response variable. The presence of an elliptical contour map suggests a notable interaction between any two factors involved in the extraction process of kajiichigoside F1 and rosamultin from the root of *R. laevigata*. Specifically, the response surface for extraction time and ethanol volume fraction was notably steep, indicating a strong interaction, whereas the response surface for the solid–liquid ratio and extraction time was relatively gentle, suggesting a weaker interaction.

### 2.3. An Evaluation of the Optimal Extraction Methodology and Its Subsequent Validation

Using Design-Expert 8.0.6 software, the optimal parameters for the ultrasonic extraction of kajiichigoside F1 and rosamultin from the root of *R. laevigata* were identified as follows: 80.0% ethanol concentration, a solid–liquid ratio of 1:25 (g/mL), an extraction duration of 80.34 min, and three extraction cycles. These parameters were subsequently refined, based on specific situational requirements, to 80.0% ethanol, a solid–liquid ratio of 1:25 (g/mL), an adjusted extraction time of 80 min, and three extraction cycles, with the procedure repeated three times for verification. The verification results demonstrated that the concentration of kajiichigoside F1 was 3.92 mg/g, while that of rosamultin was 4.99 mg/g, resulting in a total concentration of 8.91 mg/g, with a relative standard deviation (RSD) of 0.73%. These results indicate that the extraction process conditions are stable and affirm the reliability of the model’s predictions.

### 2.4. Identification of Compound Structures

The structure of the monomer compound was elucidated through its physical and chemical properties using advanced spectroscopic techniques, including ^13^C-NMR, ^1^H-NMR.


Compound **1**

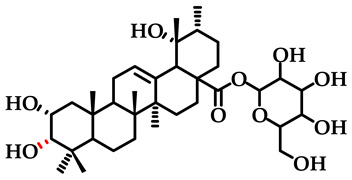



The compound, characterized as a white powder with the molecular formula C_36_H_58_O_10_, exhibited an ESI-MS m/z of 673 [M+Na]^+^. The Liebermann–Burchard reaction yielded a positive result, and the application of a 10% concentrated sulfuric acid–ethanol solution resulted in a purplish-red coloration.

In the ^1^H-NMR spectrum (400 MHz, Methanol-*d*4), the high-field region exhibits seven methyl hydrogen signals at δ: 0.90 (3H, s), 1.03 (3H, s), 1.05 (3H, d, *J* = 6.6 Hz), 1.20 (3H, s), 1.24 (3H, s), 1.37 (3H, s), and 1.59 (3H, s). The presence of a doublet at δ 1.05 suggests that the compound is likely a 19-hydroxy substituted ursane derivative. The hydrogen signal δ3.74 (1H, d, *J* = 2.5 Hz, H-3) on the carbon monoxide is given in the low field region, and it is presumed to be *α*-OH. In the low-field region, a sugar-terminal hydrogen signal is observed at δ 6.28 (1H, d, *J* = 8.4 Hz), indicating that the compound is a ursane-type monoglycoside. Additionally, the olefinic hydrogen signal is identified at δ 5.53 (1H, br s).

In the ^13^C-NMR spectrum (100 MHz, Methanol-*d*4), seven signals corresponding to methyl carbons were observed at δ: 29.4, 22.5, 17.0, 17.8, 25.2, 28.6, and 17.4 ppm. The characteristic carbon signals for the ursane C-12 and C-13 double bond appeared at δ: 124.8 ppm (C-12) and 144.4 ppm (C-13). Hydroxyl-substituted tertiary carbon signals were detected at δ: 66.4 ppm (C-2) and 78.8 ppm (C-3), while hydroxyl-substituted quaternary carbon signals were observed at δ: 73.0 ppm (C-19). The ester carbonyl carbon signal was identified at δ: 178.6 ppm (C-28). Additionally, the carbon signals for the *β*-D-glucopyranose moiety were recorded at δ: 95.8, 73.9, 78.4, 71.1, 78.7, and 62.4 ppm. Based on this data (as shown in Table 3), it was hypothesized that the compound is 2*α*, 3*α*, 19*α*-trihydroxyurs-12-en-28-oic-28-O-*β*-D-glucopyranose. Comparison of the carbon and hydrogen spectral data with the literature [18] revealed a high degree of similarity, leading to the identification of the compound as kajiichigoside F1.


Compound **2**

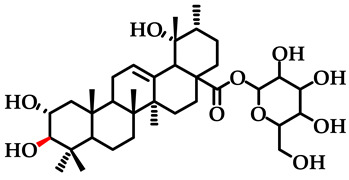



The compound, characterized as a white powder with the molecular formula C_36_H_58_O_10_, exhibited an electrospray ionization mass spectrometry (ESI-MS) *m/z* value of 673 [M+Na]^+^. The Liebermann–Burchard reaction yielded a positive result, and the application of a 10% concentrated sulfuric acid–ethanol solution produced a purplish-red coloration.

In the ^1^H-NMR (400 MHz, Methanol-*d*4) spectrum, the high field region gives seven methyl signals at δ: 0.78 (3H, s), 0.81 (3H, s), 0.93 (3H, d, *J* = 6.6 Hz), 1.01 (3H, s), 1.02 (3H, s), 1.20 (3H, s), 1.33 (3H, s), suggesting that the compound is a ursane-type compound. The hydrogen signal δ 3.42 (1H, d, *J* = 9.4 Hz, H-3) on the carbon monoxide is given in the low field region, which is presumed to be *β*-OH. A hydrogen signal δ 5.31 (1H, br s) is given in the low field region. One sugar-terminal hydrogen signal δ 5.32 (1H, d, *J* = 8.4 Hz) speculate that the compound is a ursane-type monoglycoside compound. speculate

In ^13^C-NMR (100 MHz, Methanol-*d*4) are seven methyl carbon signals δ: 29.3, 17.5, 16.6, 17.7, 24.7, 27.1, 17.1, ursane-type C-12 and C-13 double bond characteristic carbon signals δ:129.5 (C-12), 139.7 (C-13), hydroxyl-substituted tertiary carbon signals δ: 69.6 (C-2), 84.6 (C-3), hydroxyl-substituted quaternary carbon signals δ: 73.6 (C-19), ester carbonyl carbon signals δ: 178.5 (C-28). Based on this data (as shown in Table 3), carbon signal δ of a group of *β*-D-glucopyranose: 95.8,73.9, 78.3, 71.2, 78.6, 62.5. In summary, it is speculated that the compound is 2*α*, 3*β*, 19*α*-trihydroxyurs-12-en-28-oic-28-*O*-*β*-D-glucopyranose. Comparison of the carbon and hydrogen spectral data with the literature [19] revealed a high degree of similarity, leading to the identification of the compound as rosamultin.

### 2.5. Lung Index Determination Results

As illustrated in Figure 8, following stimulation with LPS, the lung index in the BALB/c mice model group was significantly elevated compared to the normal control group, indicating the presence of pulmonary edema and successful model replication. In contrast, the lung index was significantly reduced in the dexamethasone group, as well as in both the high- and low-dose groups of kajiichigoside F1, and the high-, medium-, and low-dose groups of rosamultin, when compared to the model group. Notably, no statistical difference was observed between the middle-dose group of kajiichigoside F1 and the model group. Furthermore, when compared to the normal group, no significant differences were detected in the dexamethasone group, the low- and high-dose groups of kajiichigoside F1, and the middle- and high-dose groups of rosamultin. However, significant differences were observed between the middle-dose group of kajiichigoside F1 and the low-dose group of rosamultin (*p* < 0.01 or *p* < 0.001).

### 2.6. Effect on the Pathomorphology of Lung Tissue in LPS-Induced ALI Mice

In the normal group, the lung tissue exhibited typical structural characteristics, with no significant infiltration of inflammatory cells or lymphocytes in the alveolar spaces, and the alveolar septa were slender, uniformly distributed, and structurally intact. In contrast, the model group demonstrated thickening of the lung walls, substantial infiltration of inflammatory cells and lymphocytes within the alveolar spaces and lung interstitium, altered alveolar morphology, rupture of some alveoli, and compromised alveolar structural integrity, indicating the successful establishment of an acute lung injury model. Compared to the model group, the dexamethasone group showed a marked reduction in pulmonary septal thickness, significantly decreased infiltration of inflammatory cells and lymphocytes in the alveolar spaces and pulmonary interstitium, and preservation of alveolar structural integrity. In comparison to the model group, the low-, medium-, and high-dose groups treated with kajiichigoside F1 and rosamultin exhibited significant improvements in the extent of lung tissue damage. There was a marked reduction in the infiltration of inflammatory cells and lymphocytes within the alveolar spaces and pulmonary interstitium. Additionally, the pulmonary septum was observed to be thinner, and the alveolar structures appeared relatively intact. These findings are illustrated in Figure 9.

### 2.7. Effect on the Content of Inflammatory Factors in Lung Tissue of LPS-Induced ALI Mice

In comparison to the normal group, the expression levels of TNF-α and IL-6 in the lung tissue of the model group were significantly elevated. Conversely, the expression of TNF-α and IL-6 was significantly reduced in the lung tissue of the high-, medium-, and low-dose groups treated with kajiichigoside F1 and rosamultin, relative to the model group. The dexamethasone group demonstrated a significant reduction in IL-6 expression in lung tissue; however, there was no significant difference in TNF-α levels when compared to the model group. These findings are illustrated in Figure 10 and Figure 11. Relative to the normal group, no significant differences were observed in TNF-α levels in the dexamethasone group, as well as in the high-, medium-, and low-dose groups of kajiichigoside F1, and the low- and medium-dose groups of rosamultin. However, a significant difference was noted in the high-dose group of rosamultin. Significant differences in IL-6 levels were observed in the dexamethasone group and across all dose groups of kajiichigoside F1 and rosamultin. These results suggest that kajiichigoside F1 and rosamultin effectively inhibit the secretion of the inflammatory cytokines TNF-α and IL-6 (*p* < 0.05, *p* < 0.05, or *p* < 0.001).

### 2.8. Effect on the Expression of NF-κB Pathway Protein in Lung Tissue of LPS-Induced ALI Mice

In comparison to the normal group, the expression levels of phosphorylated NF-κB p65 (p-NF-κB p65) and phosphorylated IκBα (p-IκBα) were significantly elevated in the lung tissue of the model group. Relative to the model group, the expression levels of p-NF-κB p65 were significantly reduced in the dexamethasone group, as well as in the high-, medium-, and low-dose groups of kajiichigoside F1 and rosamultin. Furthermore, the expression levels of p-NF-κB were notably decreased in the dexamethasone group, the medium- and high-dose groups of kajiichigoside F1, and the high-dose group of rosamultin. No significant differences were observed between the low-dose group of kajiichigoside F1 and the medium-dose group of rosamultin. When compared to the normal group, no significant differences were found in the expression levels of p-NF-κB p65 in the dexamethasone group, across all dose groups of kajiichigoside F1, and in the high- and low-dose groups of rosamultin. Additionally, there were no significant differences in the expression levels of p-IκBα between the dexamethasone group, the medium- and high-dose groups of kajiichigoside F1, and the medium- and high-dose groups of rosamultin. However, significant differences were noted between the low-dose groups of kajiichigoside F1 and rosamultin (*p* < 0.01 or *p* < 0.001). The findings are presented in Figure 12 and Figure 13. In summary, kajiichigoside F1 and rosamultin appear to mitigate the inflammatory response in lung tissue of mice with lipopolysaccharide (LPS)-induced acute lung injury (ALI) to a certain extent. This effect may be attributed to the inhibition of inflammatory signaling pathway activation.

## 3. Discussion

The root of *R. laevigata* is a significant medicinal resource in southern China, characterized by its extensive distribution and abundance. It serves as a crucial raw material for several Chinese patent medicines, including Sanjin Tablets, Fuke Qianjin Tablets, and Jinji Capsules. Research has identified echinacoside and roseoside as the representative active components within its triterpenoid saponins [20,21]. Our research group has been investigating the Cherokee rose root for nearly two decades [22]. In 2008, rosaponin was reported, and subsequent studies have extensively explored its pharmacological activities. These investigations have revealed that rosaponin holds potential for development as a lead compound.

The response surface optimization method is an integrative approach that combines mathematical statistics, experimental design, and optimization techniques [23]. It is extensively employed in the optimization of extraction processes [24,25]. In this study, the Box–Behnken response surface methodology was utilized to examine five factors: extraction method, extraction times, extraction duration, solid–liquid ratio, and ethanol volume fraction. This represents the first application of this method to these variables. The concentrations of kajiichigoside F1 and rosamultin were used as evaluation indices. The impact of each factor on the response value was assessed more clearly and intuitively through three-dimensional graphical representations. By developing a model, fitting a functional equation, and analyzing the relationship between the response value and each factor, the optimal preparation process was determined to be as follows: ultrasonic extraction with 80.0% ethanol for 80 min, repeated three times, with a solid–liquid ratio of 1:25. This study constitutes the first report on the optimization of the extraction process for kajiichigoside F1 and rosamultin from the root of *R. laevigata*, thereby addressing a gap in the large-scale preparation of these two saponins. The optimized process is anticipated to facilitate large-scale production of kajiichigoside F1 and rosamultin, thereby reducing costs, enhancing efficiency, and increasing application value.

Kajiichigoside F1 and rosamultin, found in the root of *R. laevigata*, are ursane-type pentacyclic triterpenoid saponins and represent a pair of epimers differing only in the configuration of the 3-hydroxyl substitution. Specifically, kajiichigoside F1 is characterized by the α configuration, whereas rosamultin is characterized by the β configuration at this position. These subtle differences in substituents result in distinct pharmacological effects between kajiichigoside F1 and rosamultin. It has been documented that both compounds possess certain anti-inflammatory activities and can effectively inhibit nitric oxide (NO) production, although the precise mechanisms remain unclear [26]. There is a known association between oxidative stress injury and inflammation, with excessive oxidative stress promoting inflammatory processes [27]. Research indicates that rosamultin may have therapeutic effects on oxidative stress injury [13], although its anti-inflammatory properties require further investigation. Additionally, kajiichigoside F1 shows potential in neuroprotection by activating the AMPAR and BDNF/AKT/mTOR signaling pathways [28]. The signaling pathways activated by the anti-hypoxia activity of kajiichigoside F1 and rosamultin were inconsistent. Kajiichigoside F1 protected vascular endothelial cells from hypoxia-induced mitochondrial apoptosis by activating ERK1/2 signaling. Rosamultin protects vascular endothelial cells from hypoxia-induced apoptosis by activating the PI3K/AKT signaling pathway [12]. In addition, rosamultin plays an anti-radiation protective role by promoting the Nrf2/HO-1 signaling pathway and reducing radiation-induced oxidative stress [29]. Rosamultin can improve cisplatin-induced renal histopathological damage and fibrosis by inhibiting the activation of p38 and JNK [30]. Rosamultin significantly mitigated oxidative stress and inflammation induced by HH, enhanced abnormal bone metabolism, and ameliorated the imbalance in bone remodeling in rats by inhibiting the expression of sclerostin and activating the Wnt/β-catenin signaling pathway [14]. Despite this, there is a paucity of studies on the anti-inflammatory properties of kajiichigoside F1 and rosamultin. Consequently, a mouse model of acute lung injury was developed through intratracheal administration of lipopolysaccharide (LPS) [31] to examine the anti-inflammatory protective effects of kajiichigoside F1 and rosamultin against LPS-induced acute lung injury in mice.

Acute lung injury is characterized by an inflammatory response triggered by lipopolysaccharide (LPS) stimulation, which is associated with the activation of nuclear factor kappa B (NF-κB) and the secretion of pro-inflammatory cytokines, including tumor necrosis factor-alpha (TNF-α) and interleukin-6 (IL-6) [32,33]. NF-κB serves as a pivotal transcription factor in the regulation of inflammatory responses. Upon LPS stimulation, a cascade of events is initiated, beginning with the activation of IκB kinase (IKK). This activation leads to the phosphorylation of serine residues at the regulatory sites of the IκB subunits IKKα and IKKβ within the cell. Consequently, the IκB subunit undergoes ubiquitination and subsequent proteolytic degradation, resulting in the release of NF-κB. Concurrently, the expression of the IκBα gene is activated, leading to the phosphorylation of IκBα and the inhibition of NF-κB activity. The liberated NF-κB translocates to the nucleus, where it binds to target genes containing NF-κB binding sites, thereby initiating transcription. This process induces the production of pro-inflammatory cytokines such as TNF-α and IL-6, thereby amplifying the inflammatory response [34]. This exacerbated inflammatory activity contributes to the stimulation of inflammation within lung tissue, the disruption of pulmonary microvascular endothelial cells, and the damage to lung epithelial cells, ultimately culminating in acute lung injury [35]. Consequently, suppressing the expression of these pro-inflammatory mediators and cytokines may effectively ameliorate acute lung injury.

This study provides the first evidence that kajiichigoside F1 and rosamultin exert a protective effect against lipopolysaccharide (LPS)-induced acute lung injury in mice. The findings suggest that these compounds may inhibit the NF-κB signaling pathway by attenuating pathological changes in lung tissue, decreasing the levels of inflammatory cytokines TNF-α and IL-6, and reducing the expression of NF-κB p65 and phosphorylated IκBα proteins. Consequently, kajiichigoside F1 and rosamultin demonstrate potential anti-inflammatory activity.

## 4. Materials and Methods

### 4.1. Single Factor Experiment

In accordance with the traditional single-factor experimental approach [36], the following parameters were systematically examined: (1) Extraction method: Three samples of medicinal powder of *R. laevigata*, each precisely weighing approximately 2 g, were utilized. The solvent employed was 60% ethanol, with a material-to-liquid ratio of 1:10 and an extraction duration of 40 min. The extraction techniques applied included ultrasonic extraction, heating reflux, and water decoction. Subsequently, the extraction yields of kajiichigoside F1 and rosamultin were quantitatively assessed. Three parallel samples were prepared, and the mean value was calculated. (2) Ethanol volume fraction: Three samples of medicinal powder of *R. laevigata*, each precisely weighing approximately 2 g, were prepared with a material-to-liquid ratio of 1:10. Each sample underwent a single extraction process lasting 40 min. The ethanol volume fractions employed were 20%, 40%, 60%, 80%, and 95%. Subsequently, the extraction yields of kajiichigoside F1 and rosamultin were measured. The procedure was conducted in triplicate for each condition, and the results were averaged. (3) Solid–liquid ratio: Three samples of medicinal powder of *R. laevigata*, each weighing approximately 2 g, were subjected to ultrasonic extraction using 80% ethanol. The extraction process was conducted for 40 min at varying solid–liquid ratios of 1:10, 1:15, 1:20, 1:25, and 1:30 g/mL. Subsequently, the extraction yields of kajiichigoside F1 and rosamultin were quantified. Each experimental condition was performed in triplicate, and the results were averaged to ensure accuracy and reliability. (4) Extraction time: Three samples of medicinal powder of *R. laevigata*, each precisely weighing approximately 2 g, were subjected to ultrasonic extraction using 80% ethanol. The material-to-solvent ratio was maintained at 1:20 g/mL. The extraction process was conducted for varying durations of 40, 60, 80, 100, and 120 min. Following extraction, the yields of kajiichigoside F1 and rosamultin were quantified. Each experimental condition was performed in triplicate, and the resulting data were averaged for analysis. (5) Three precisely weighed portions of medicinal powder of *R. laevigata*, each approximately 2 g, were utilized. The material-to-liquid ratio was maintained at 1:20 g/mL, employing 80% ethanol for ultrasonic extraction over a duration of 100 min. The extraction process was conducted one, two, and three times. Subsequently, the extraction yields of kajiichigoside F1 and rosamultin were quantified. Each experimental condition was performed in triplicate, and the results were averaged. In examining the impact of an individual factor on the extraction rate of kajiichigoside F1 and rosamultin, the intermediate levels of other factors were maintained constantly. The yield of kajiichigoside F1 and rosamultin served as the evaluation metric in this study, facilitating the determination of the optimal range for each factor.

### 4.2. Box–Behnken Response Surface Experiment

The response surface design was conducted using Design-Expert 8.0.6 software. Based on the results of a single-factor test, the Box–Behnken response surface methodology was employed to design an experiment involving four factors at three levels: ethanol volume fraction of the extraction solvent (A), solid–liquid ratio (B), extraction time (C), and number of extraction cycles (D). The response variables were the extraction rates of kajiichigoside F1 and rosamultin, as well as their combined total extraction rate from the root of *R. laevigata*. The levels of the factors were coded as −1, 0, and +1, resulting in a total of 29 experimental runs.

### 4.3. Preparation of Kajiichigoside F1 and Rosamultin

Based on the optimal extraction conditions determined through Box–Behnken response surface methodology, 20 kg of *R. laevigata* root, sourced from Guangxi Xianzhu Traditional Chinese Medicine Science and Technology Co., Ltd. in Nanning, China, was subjected to ultrasonic extraction using 80% ethanol. This process was repeated three times, each lasting 80 min, with an herb-to-solvent ratio of 1:25 g/mL. The organic solvent was subsequently recovered under reduced pressure to yield the total extract. A portion, constituting one-tenth of the total extract, was further processed using dichloromethane obtained from Chengdu Cologne Chemical Co., Ltd., Chengdu, China. The saponin components were concentrated using a D101 macroporous resin column from Tianjin Guangfu Fine Chemical Research Institute, Tianjin, China, and subsequently separated using 200–300 mesh silica gel column chromatography from Sinopharm Group Reagent Co., Ltd., Shanghai, China. The sample underwent repeated purification via preparative liquid chromatography, employing a methanol–water elution system. Chromatographic methanol was provided by Xilong Chemical Co., Ltd., Shantou, China, and laboratory ultrapure water was used. Following the recovery of the reagent using a rotary evaporator (EYELA, Tokyo, Japan) under reduced pressure, the sample was subsequently dried utilizing a freeze dryer (EYELA FDS-1000, Tokyo, Japan). The compound was then identified via nuclear magnetic resonance spectroscopy. The resultant compounds, identified as kajiichigoside F1 (1.1321 g) and rosamultin (1.2755 g), exhibited a purity exceeding 99.0%, as determined by high-performance liquid chromatography (HPLC) using a 2695 system (Waters Limited, Milford, MA, USA).

### 4.4. Structure Identification of Compounds

The structure of the monomer compound was elucidated through its physical and chemical properties using nuclear magnetic resonance spectroscopy, including ^13^C-NMR and ^1^H-NMR, along with other contemporary analytical techniques.

### 4.5. Animal Model Construction and Administration of Drugs

A total of 90 specific pathogen-free (SPF) male BALB/c mice, aged 6 weeks and weighing between 18–22 g, were procured from Hunan Slake Jinda Laboratory Animal Co., Ltd., Changsha, China (License No.: SCXK (Xiang) 2019-0004) for the purposes of this study. The mice were maintained in a controlled environment with regulated temperature and humidity and were provided with ad libitum access to food and water. Following a 7-day acclimatization period, the mice were randomly assigned to nine groups based on body weight: a normal group, a model group, a dexamethasone group (5 mg/kg), a high-dose kajiichigoside F1 group (20 mg/kg), a medium-dose kajiichigoside F1 group (10 mg/kg), a low-dose kajiichigoside F1 group (5 mg/kg), a high-dose rosamultin group (20 mg/kg), a medium-dose rosamultin group (10 mg/kg), and a low-dose rosamultin group (5 mg/kg), with each group comprising 10 mice. Mice in each experimental group were anesthetized using sodium pentobarbital and subsequently administered an intratracheal infusion of lipopolysaccharide (LPS) at a dosage of 4 mg/kg (Sigma Co., Ltd., St. Louis, MO, USA). The control group remained untreated. Three hours post-induction, the high-, medium-, and low-dose groups received intragastric administration of kajiichigoside F1 and rosamultin. Meanwhile, the control and model groups were administered an equivalent volume of normal saline, and the dexamethasone group received an intraperitoneal injection. These treatments were conducted once daily for a duration of five consecutive days. The body weights of the mice were recorded daily. All animal procedures were conducted in compliance with the ethical guidelines approved by the Animal Experimental Ethics Committee of Guangxi University of Traditional Chinese Medicine (Approval No. DW20200520-322).

### 4.6. Determination of Lung Index

The ratio of lung tissue wet mass to body mass was determined by measuring the mass of the lung tissue.

### 4.7. HE Staining to Observe Histopathological Changes in the Lung Tissue

The lung tissues were fixed in a 4% paraformaldehyde–phosphate buffer solution for a duration of 12 h, followed by embedding in standard paraffin for subsequent sectioning. Morphological changes were then examined using a conventional optical microscope after hematoxylin and eosin (HE) staining, as provided by Beijing Suolaibao Biotechnology Co., Ltd., Beijing, China.

### 4.8. ELISA Reagent for the Detection of the Expression of Inflammatory Factors in Lung Tissue

Following the homogenization of murine lung tissue, the concentrations of TNF-α and IL-6 were quantified using an ELISA kit, in accordance with the manufacturer’s protocol (Xinbosheng Biotechnology Co., Ltd., Shenzhen, China).

### 4.9. Western Blot Method

The lung tissues were homogenized, and proteins were extracted using RIPA lysis buffer. Protein concentrations were determined with the BCA assay (Thermo Fisher Scientific, Waltham, MA, USA). The samples were then boiled and denatured by adding protein loading buffer and subjected to SDS-PAGE electrophoresis (Bio-Rad Laboratories, Hercules, CA, USA). Subsequently, the proteins were transferred to a PVDF membrane (Bio-Rad Laboratories, Hercules, CA, USA). After blocking for 15 min, the membrane was washed three times with 1× TBST and incubated overnight at 4 °C with primary antibodies targeting GAPDH, NF-κB p65, p-NF-κB p65, IKKα, IKKβ, p-IKKα/β, IκBα, and p-IκBα (Cell Signaling Technology, Boston, MA, USA). The following day, the membrane was equilibrated to room temperature for 15 min, washed three times, and incubated with a secondary antibody for 60 min at room temperature. After three additional washes, the chemiluminescent signal was developed, and the gray values were quantified using ImageJ-win64 software.

### 4.10. Statistical Analysis

Statistical analysis was conducted utilizing GraphPad Prism 8.0 software, based on three independent experimental replicates. Each experiment was performed in triplicate to ensure reproducibility. Data analysis across various experiments employed the independent *t*-test and one-way ANOVA. Results are presented as mean ± standard deviation (mean ± S.D.), with a *p*-value of less than 0.05 considered indicative of statistical significance.

## 5. Conclusions

In this study, the optimal extraction process for kajiichigoside F1 and rosamultin from the root of *R. laevigata* was determined using a single-factor experiment and the Box–Behnken response surface methodology. Subsequently, an acute lung injury experiment was conducted using the extracted kajiichigoside F1 and rosamultin. The results indicated that both compounds could mitigate acute lung injury by inhibiting the activation of the NF-κB signaling pathway. These findings offer a theoretical framework for further exploration of the molecular mechanisms underlying the anti-inflammatory properties of kajiichigoside F1 and rosamultin and provide a foundational basis for their potential clinical application in treating acute lung injury in the future.

## Figures and Tables

**Figure 1 pharmaceuticals-18-00253-f001:**
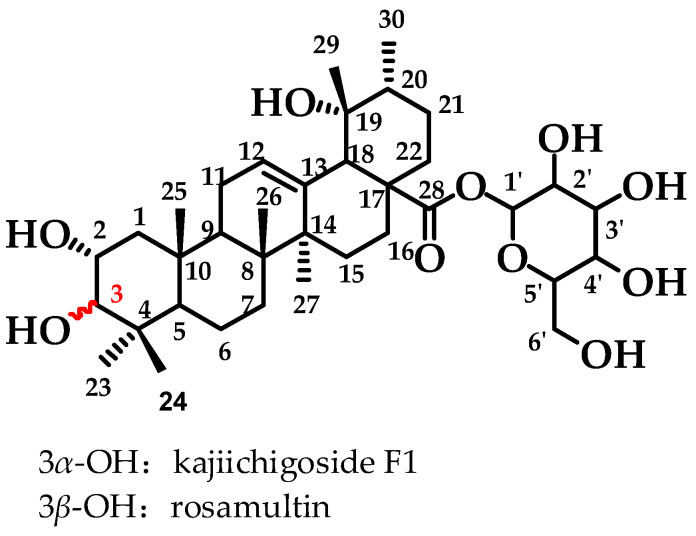
Chemical structural formula of kajiichigoside F1 and rosamultin.

**Figure 2 pharmaceuticals-18-00253-f002:**
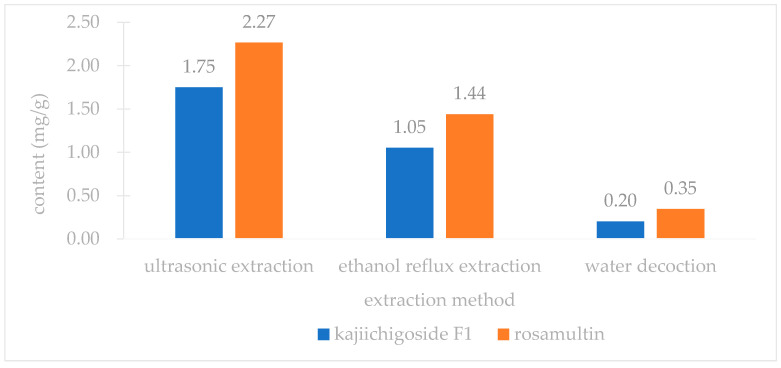
Effects of various extraction techniques on the concentrations of kajiichigoside F1 and rosamultin in the root of *R. laevigata* (*n* = 3).

**Figure 3 pharmaceuticals-18-00253-f003:**
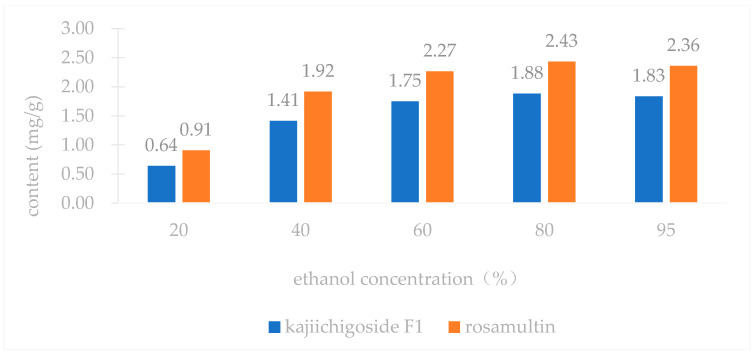
Effects of varying ethanol concentrations on the levels of kajiichigoside F1 and rosamultin in the root of *R*. *laevigata* (*n* = 3).

**Figure 4 pharmaceuticals-18-00253-f004:**
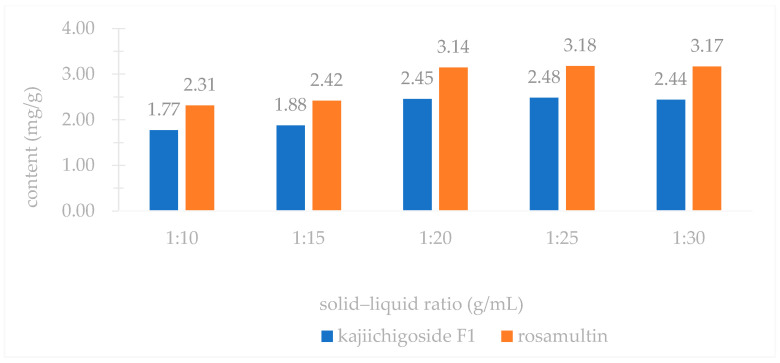
Effects of different solid–liquid ratios on the contents of kajiichigoside F1 and rosamultin in the root of *R. laevigata (n* = 3).

**Figure 5 pharmaceuticals-18-00253-f005:**
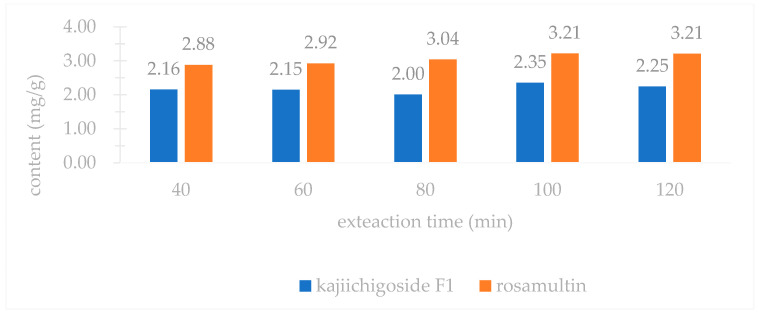
Effects of different extraction time on the content of kajiichigoside F1 and rosamultin in the root of *R. laevigata* (*n* = 3).

**Figure 6 pharmaceuticals-18-00253-f006:**
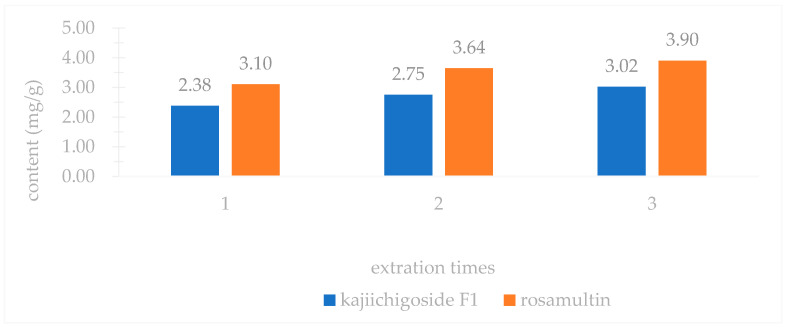
Effects of different extraction times on the content of kajiichigoside F1 and rosamultin in the root of *R. laevigata* (*n* = 3).

**Figure 7 pharmaceuticals-18-00253-f007:**
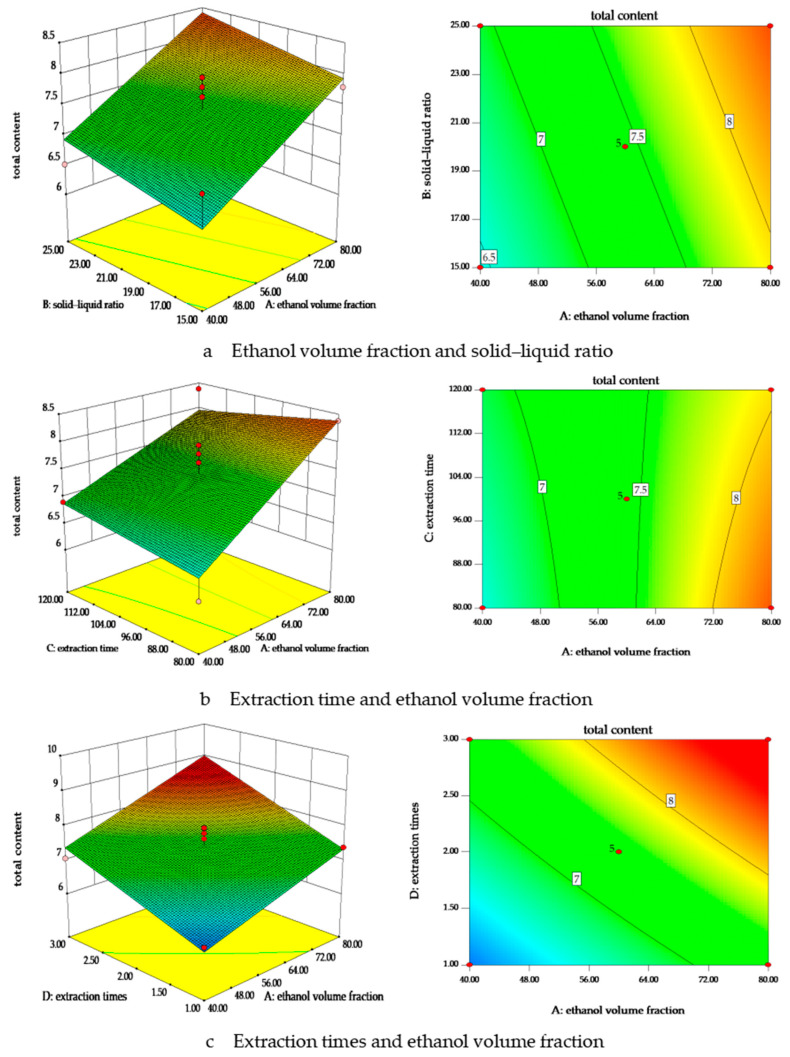
The response surface diagram illustrates the influence of various factors on the extraction of kajiichigoside F1 and rosamultin from the root of *R. laevigata*. Utilizing Design-Expert 8.0.6 software, the response surface analysis was conducted to examine the interactions among these factors, and the corresponding response surface curves were generated. A steeper response surface indicates a more pronounced effect on the extraction rate. Specifically, the diagrams depict (**a**) the impact of ethanol volume fraction and solid–liquid ratio on the total content of kajiichigoside F1 and rosamultin; (**b**) the influence of extraction time and ethanol volume fraction; (**c**) the effects of the number of extraction times and ethanol volume fraction; (**d**) the interaction between extraction time and solid–liquid ratio; (**e**) the combined effects of the number of extraction times and solid–liquid ratio; and (**f**) the relationship between the number of extraction time and extraction times on the total content of kajiichigoside F1 and rosamultin.

**Figure 8 pharmaceuticals-18-00253-f008:**
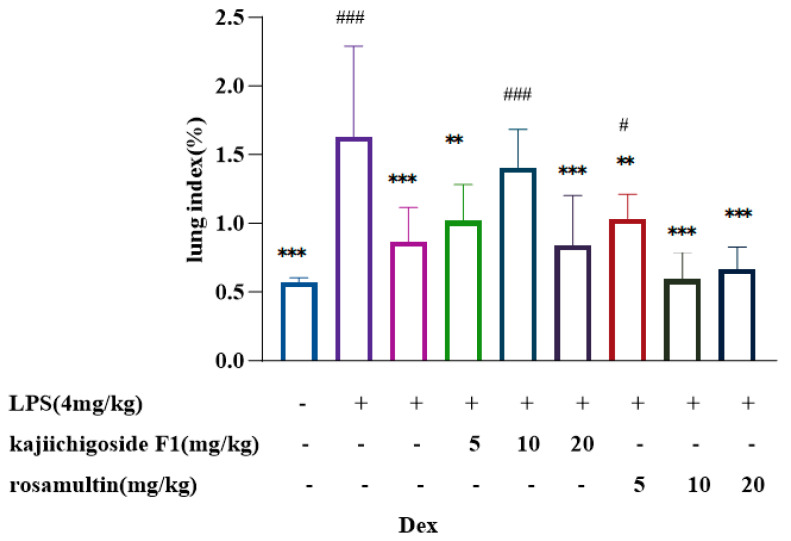
Effects of kajiichigoside F1 and rosamultin on lung index in LPS-induced ALI mice. The lung index was analyzed by calculating the wet weight/body weight ratio of lung tissue in ALI mice. Results are expressed as mean ± standard deviation (S.D.). * means compared with the model group: ** *p* < 0.01, *** *p* < 0.001; # means compared with the normal group: # *p* < 0.05, ### *p* < 0.001.

**Figure 9 pharmaceuticals-18-00253-f009:**
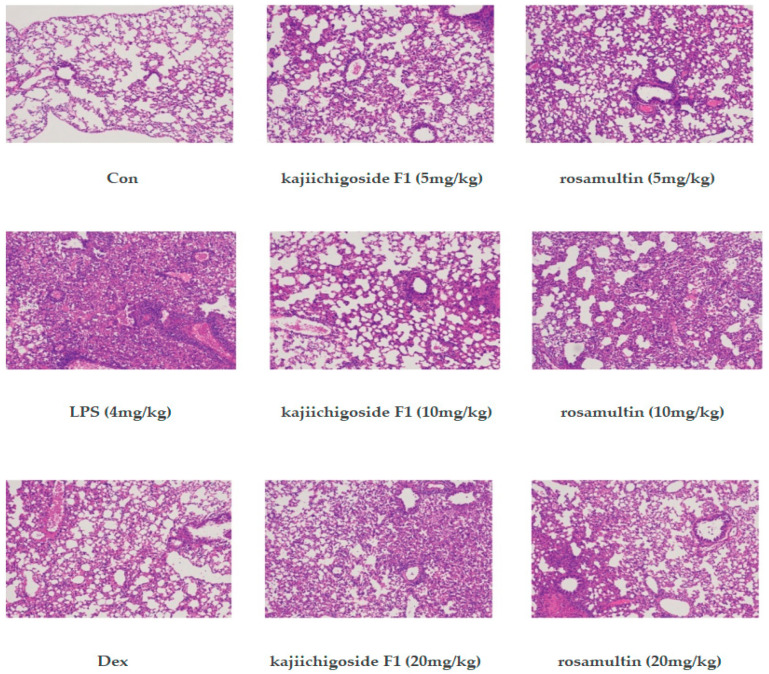
Effects of kajiichigoside F1 and rosamultin on the pathomorphology of lung tissue in LPS-induced ALI mice (HE staining, ×100). Representative images of lung tissue pathology were obtained by performing H&E staining on each group of mice. Con: normal group; LPS: model group; Dex: dexamethasone group; kajiichigoside F1: low-, medium-, and high-dose group; rosamultin: low-, medium-, and high-dose group.

**Figure 10 pharmaceuticals-18-00253-f010:**
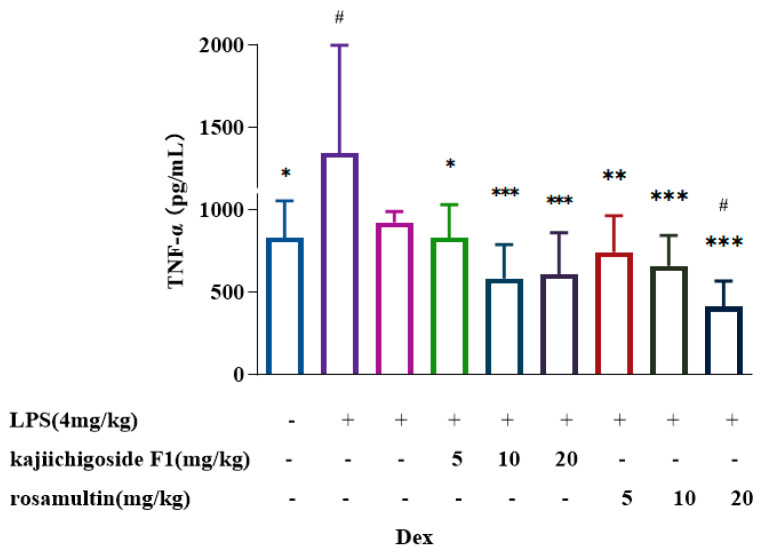
The effect of kajiichigoside F1 and rosamultin on the inhibition of pro-inflammatory cytokine TNF-α in lung tissue of LPS-induced ALI mice. Results are expressed as mean ± standard deviation (S.D.). * means compared with the model group: * *p* < 0.05, ** *p* < 0.01, *** *p* < 0.001; # means compared with the normal group: # *p* < 0.05.

**Figure 11 pharmaceuticals-18-00253-f011:**
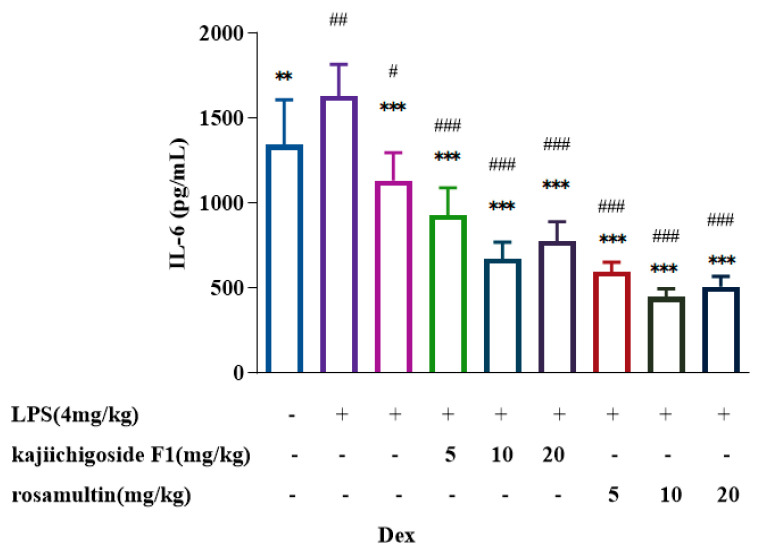
The effect of kajiichigoside F1 and rosamultin on the inhibition of pro-inflammatory cytokine IL-6 in lung tissue of LPS-induced ALI mice. Results are expressed as mean ± standard deviation (S.D.). * means compared with the model group: ** *p* < 0.01, *** *p* < 0.001; # means compared with the normal group: # *p* < 0.05, ## *p* < 0.01, ### *p* < 0.001.

**Figure 12 pharmaceuticals-18-00253-f012:**
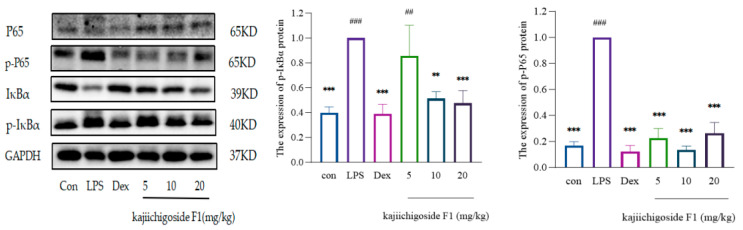
The inhibitory effect of kajiichigoside F1 on the expression of NF-κB pathway protein in LPS-induced ALI mice. Results are expressed as mean ± standard deviation (S.D.). * means compared with the model group: ** *p* < 0.01, *** *p* < 0.001; # means compared with the normal group: ## *p* < 0.01, ### *p* < 0.001.

**Figure 13 pharmaceuticals-18-00253-f013:**
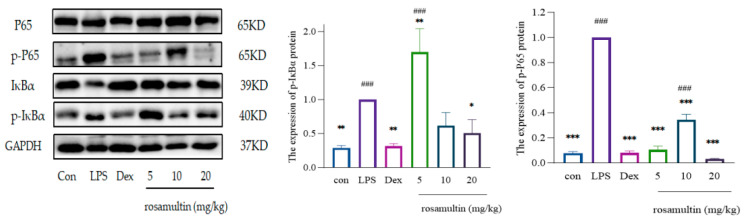
The inhibitory effect of rosamultin on the expression of NF-κB pathway proteins in LPS-induced ALI mice. Results are expressed as mean ± standard deviation (S.D.). * means compared with the model group: * *p* < 0.05, ** *p* < 0.01, *** *p* < 0.001; # means compared with the normal group: ### *p* < 0.001.

**Table 1 pharmaceuticals-18-00253-t001:** Box–Behnken experimental design and outcomes for extracting kajiichigoside F1 and rosamultin from the root of *R. laevigata*.

Test Number	A (%)	B (g/mL)	C (min)	D (Times)	Measured Extraction Rate (mg/g)		
					Kajiichigoside F1	Rosamultinin	Total Content
1	40 (−1)	20 (0)	80 (−1)	2	2.6970	3.3980	6.0950
2	60 (0)	20	80	1 (−1)	3.0546	3.7447	6.7993
3	80 (+10)	20	120 (+1)	2 (0)	3.8276	4.5521	8.3797
4	60	20	100 (0)	2	3.5124	4.2764	7.7889
5	60	15 (−1)	120	2	3.4672	4.2044	7.6716
6	60	20	100	2	3.4277	4.2007	7.6284
7	60	15	100	1	2.5424	3.0978	5.6402
8	40	20	120	2	3.0748	3.8292	6.9041
9	60	20	120	1	2.7575	3.3597	6.1173
10	80	20	100	3 (+1)	3.9361	4.6863	8.6224
11	80	20	100	1	3.3322	4.0450	7.3772
12	40	20	100	3	3.1416	3.9086	7.0503
13	60	15	100	3	3.4384	4.1653	7.6037
14	60	20	100	2	3.1751	3.8574	7.0326
15	60	15	80	2	3.3843	4.1233	7.5076
16	60	25 (+1)	100	3	3.7045	4.5394	8.2439
17	80	20	80	2	3.8197	4.5587	8.3784
18	60	25	100	1	3.0029	3.6433	6.6462
19	40	20	100	1	2.7268	3.3923	6.1191
20	60	20	120	3	3.6350	4.3975	8.0325
21	60	25	120	2	3.7158	4.5409	8.2566
22	60	20	100	2	3.7083	4.4912	8.1995
23	80	25	100	2	3.6813	4.3265	8.0078
24	40	25	100	2	2.9064	3.6017	6.5081
25	60	20	80	3	3.7359	4.5478	8.2836
26	80	15	100	2	3.5781	4.2103	7.7884
27	60	20	100	2	3.5965	4.3477	7.9442
28	60	25	80	2	3.7914	4.6303	8.4216
29	40	15	100	2	3.1011	3.8798	6.9809

**Table 2 pharmaceuticals-18-00253-t002:** A variance analysis of the test results from the extraction process of kajiichigoside F1 and rosamultin in the root of *R. laevigata*.

Source of Variance	Quadratic Sum	Degree of Freedom	Mean Square	F Ratio	*p* Ratio
model	16.34	14	1.17	5.54	0.0014
A	6.60	1	6.60	31.30	<0.0001
B	0.70	1	0.70	3.31	0.0904
C	0.00	1	0.00	0.01	0.9391
D	6.96	1	6.96	33.01	<0.0001
AB	0.12	1	0.12	0.57	0.4634
AC	0.16	1	0.16	0.77	0.3938
AD	0.02	1	0.02	0.12	0.7374
BC	0.03	1	0.03	0.13	0.7254
BD	0.03	1	0.03	0.16	0.6964
CD	0.05	1	0.05	0.22	0.6461
A^2^	0.33	1	0.33	1.57	0.2303
B^2^	0.06	1	0.06	0.27	0.6143
C^2^	0.07	1	0.07	0.33	0.5752
D^2^	1.23	1	1.23	5.84	0.0299
residual error	2.95	14	0.21		
lack of fit	2.18	10	0.22	1.14	0.4887
error	0.77	4	0.19		
total deviation	19.29	28			

**Table 3 pharmaceuticals-18-00253-t003:** ^13^C-NMR data of kajiichigoside F1 and rosamultin.

Kajiichigoside F1		Rosamultin	
Position	Experimental Value	Position	Experimental Value
C-1	42.7	C-1	48.2
C-2	66.4	C-2	69.6
C-3	78.8	C-3	84.6
C-4	39.4	C-4	39.2
C-5	48.1	C-5	56.7
C-6	19.7	C-6	19.7
C-7	33.8	C-7	34.1
C-8	40.9	C-8	41.3
C-9	47.1	C-9	48.6
C-10	40.5	C-10	40.5
C-11	25.0	C-11	24.8
C-12	124.8	C-12	129.5
C-13	144.4	C-13	139.7
C-14	42.7	C-14	42.7
C-15	29.3	C-15	29.6
C-16	26.5	C-16	26.5
C-17	48.1	C-17	49.4
C-18	55.0	C-18	55.0
C-19	73.0	C-19	73.6
C-20	41.5	C-20	42.9
C-21	28.5	C-21	27.2
C-22	39.4	C-22	38.3
C-23	29.4	C-23	29.3
C-24	22.5	C-24	17.5
C-25	17.0	C-25	16.6
C-26	17.8	C-26	17.7
C-27	25.2	C-27	24.7
C-28	178.6	C-28	178.5
C-29	28.5	C-29	27.1
C-30	17.4	C-30	17.1
C-1’	95.8	C-1’	95.8
C-2’	73.9	C-2’	73.9
C-3’	78.4	C-3’	78.3
C-4’	71.1	C-4’	71.2
C-5’	78.7	C-5’	78.6
C-6’	62.4	C-6’	62.5

## Data Availability

Data are contained within the article.

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
