# Peer review of "Study on the Optimization of an Extraction Process of Two Triterpenoid Saponins in the Root of Rosa laevigata Michx. and Their Protective Effect on Acute Lung Injury"

_pharmaceuticals, 2025, doi:10.3390/ph18020253_

Round 1

Reviewer 1 Report

Comments and Suggestions for Authors

Dear authors

The MS entitled “Study on the Optimization of Preparation Process of Two 2 Triterpenoid Saponins in root of Rosa laevigata Michx. and Theirs Protective Effect on Acute Lung Injury”.  The aim of this study was to establish the exact mechanism of anti-inflammatory activities of two isolated saponins enantiomers (kajiichigoside F1 and rosamultin ) from the plant Rosa laevigata. Also, the authors performed RSM on various factors to optimize the extraction process for better yield. The anti-inflammatory mechanism was determined using LPS-induced acute lung injury model. The data is OK and MS is fine. My comments and suggestions are:

·       In my opinion the title should replace “preparation” by “extraction while Correct “theirs” should be corrected as “their”

·       Line 15. Delete the line “it has been reported that”.

·       What’s meant by “preparation process”?

·       Line 47-50, correct the sentences.

·       Line 83-86, refine the sentences.

·       R. laevigata should be italic through out the MS.

·       Line 188. How the alpha and beta position of OH at position 3 was established? Was there any significant difference between both in their physical/chemical properties?

·       Provide H-NMR and 13C-NMR spectra of both the compounds.

Comments on the Quality of English Language

English language should be revised.  

Author Response

Thank you sincerely for your insightful comments and professional guidance. Your feedback has been instrumental in enhancing the academic rigor of our manuscript. In response to your suggestions and requests, we have made the necessary revisions to the manuscript. We are hopeful that these changes have further improved the quality of our work. Please find below our detailed response:

Response to Expert 1:

Thank you for your valuable suggestions. I will carefully consider your feedback to make the necessary revisions to the manuscript. Below are my responses to your comments:

  1. In my opinion the title should replace “preparation” by “extraction while Correct “theirs” should be corrected as “their”

Answer : I have replaced the term "preparation" with "extraction" and corrected "theirs" to "their."

  1. Line 15. Delete the line “it has been reported that”.

Answer : I have removed the phrase "it has been reported that."

  1. What’s meant by “preparation process”?

Answer : The phrase "preparation process" may have been inaccurately translated by the author, and I will ensure its proper contextual use.

  1. Line 47-50, correct the sentences.

Answer : The sentences located in lines 47-50 have been revised. The pathogenesis of acute lung injury (ALI) is predominantly linked to pulmonary edema, inflammatory infiltration, and damage to the pulmonary microvascular endothelial cells. Consequently, a pivotal therapeutic strategy for ALI entails the precise modulation of the inflammatory response.

  1. Line 83-86, refine the sentences.

Answer: The sentences located in lines 83-86 have been omitted. Importantly, administration of kajiichigoside F1 and rosamultin led to a marked decrease in the expression levels of tumor necrosis factor-alpha (TNF-α) and interleukin-6 (IL-6), as well as in the phosphorylation levels of nuclear factor kappa-light-chain-enhancer of activated B cells (NF-κB) p65 and inhibitor of kappa B alpha (IκBα). These findings highlight the protective effects of kajiichigoside F1 and rosamultin against lipopolysaccharide (LPS)-induced acute lung injury and elucidate their underlying mechanisms of action, thereby establishing a foundational basis for their further development.

  1. laevigata should be italic through out the MS.

Answer: The term "R. laevigata" was initially meant to serve as an abbreviation for the root of Rosa laevigata. However, it was inadvertently not formatted in italics, a mistake that has since been rectified to "R. laevigata."

  1. Line 188. How the alpha and beta position of OH at position 3 was established? Was there any significant difference between both in their physical/chemical properties?

Answer: The stereochemical configuration at the C-3 position is α-OH and β-OH, respectively, and these configurations are epimeric to each other. Drawing on previous literature, the coupling constants in the proton nuclear magnetic resonance (1H-NMR) spectrum and the chemical shift values in the carbon nuclear magnetic resonance (13C-NMR) spectrum are elucidated. The primary distinctions observed in the NMR spectra are as follows: In the 1H-NMR spectrum, (1) when the C-3 position is α-OH, as in echinacoside, the coupling constant (J) for H-3 ranges from 2.0 to 5.0 Hz; conversely, when the C-3 position is β-OH, the coupling constant (J) for H-3 ranges from 6.0 to 10.0 Hz. (2) There is no significant difference in the chemical shift of the hydrogen spectrum between the two epimers. In 13C-NMR spectroscopy, the chemical shift differences at the C-3 position between the two compounds are notably significant, with variations also observed at the C-1, C-2, and C-5 positions. Specifically, when the hydroxyl group at the C-3 position is in the α configuration, the chemical shifts are δ76.0-79.9 ppm for C-3, δ41.0-43.0 ppm for C-1, δ64.0-67.0 ppm for C-2, and δ48.0-49.9 ppm for C-5. Conversely, when the hydroxyl group at the C-3 position is in the β configuration, the chemical shifts are δ81.0-86.0 ppm for C-3, δ48.0-49.9 ppm for C-1, δ68.0-69.9 ppm for C-2, and δ55.0-56.9 ppm for C-5.

Reviewer 2 Report

Comments and Suggestions for Authors

The manuscript is interesting but not well written. There are too many errors of language, making it quite difficult to read in some places. It is also quite verbose and unclear, not concisely organized. It should be revised with the help of native-English professional.

It would be useful to indicate the chemical structure of kaji-ichigoside F1 and rosamultinin at the beginning of the manuscript, to facilitate its understanding, with a single formula for the two isomers.

Figure 6 with the multiple response surface diagrams is graphically nice but useless if the different panels are not explained in the text. The crude results are given but not sufficiently explained in a textual form. The explanation text should be significantly improved.

The discussion is not well organized. It would be clearer to discuss first the isolation process, and then the compound properties and pharmacology.

Previous studies with kaji-ichigoside F1 and rosamultinin should be further considered. The two compounds have revealed a myriad of activities, affecting different inflammatory pathways. The scheme in Fig 13 (which is not a “technical route”) is far too restrictive. This naïve Figure is incorrect and should be removed.

It would be useful to add a conclusion at the end of the manuscript.

The manuscript is not well adapted to the journal. There is no pharmaceutical aspect. It would be more appropriate in Molecules (mdpi), for example.

Other points

-        In the title, « Rosa laevigata » should be in italic and throughout the manuscript.

-        Abstract: “the content of kajiichigoside F1 and rosamultin and the total content of the two saponins” useless duplicated term.

-        Introduction: “Our research group began to study…” if a previous work has been done, indicate the reference.

-        Compounds should be defined properly and in a homogeneous form. kaji-ichigoside or kajiichigoside ? rosa-multinin (page 9) or rosamultin or rosamultinin ? The present mixture is not acceptable.

-        Provide complete analytical data for the two compounds in the Materials & Methods section (not in the main text).

Comments on the Quality of English Language

The English language and organization of the paper should be reworked.

Author Response

Thank you sincerely for your insightful comments and professional guidance. Your feedback has been instrumental in enhancing the academic rigor of our manuscript. In response to your suggestions and requests, we have made the necessary revisions to the manuscript. We are hopeful that these changes have further improved the quality of our work. Please find below our detailed response:

Response to Expert 2:

Thank you for your valuable suggestions. I will carefully consider your feedback to make the necessary revisions to the manuscript. Below are my responses to your comments:

  1. It would be useful to indicate the chemical structure of kaji-ichigoside F1 and rosamultinin at the beginning of the manuscript, to facilitate its understanding, with a single formula for the two isomers.

Answer: The chemical structural formulas of echinacoside and roseoside are presented in the "Introduction" section.

kajiichigoside F1.:        rosamultin:

  1. Figure 6 with the multiple response surface diagrams is graphically nice but useless if the different panels are not explained in the text. The crude results are given but not sufficiently explained in a textual form. The explanation text should be significantly improved.

Answer: The response surface methodology diagram has been enhanced. The analysis of interactions among various factors was performed utilizing Design Expert 8.0.6 software, enabling the creation of response surface plots and contour maps. These visualizations, presented in Figure 7, effectively illustrate the interactions among the experimental factors. The elliptical contour map indicates a significant interaction between the factors involved in the extraction process of kajiichigoside F1 and rosamultin from the root of R. laevigata. Notably, the response surface for extraction time and ethanol volume fraction exhibits a pronounced steepness, reflecting a strong interaction. In contrast, the response surface for the solid-liquid ratio and extraction time is comparatively gentle, implying a weaker interaction.

  1. The discussion is not well organized. It would be clearer to discuss first the isolation process, and then the compound properties and pharmacology.

Answer: The discussion section has been restructured to first address the medicinal materials, extraction process, monomer properties, and pharmacological effects.

  1. Previous studies with kaji-ichigoside F1 and rosamultinin should be further considered. The two compounds have revealed a myriad of activities, affecting different inflammatory pathways. The scheme in Fig 13 (which is not a “technical route”) is far too restrictive. This naïve Figure is incorrect and should be removed.

Answer: The study identified that Kajiichigoside F1 exhibits anti-hypoxia activity through the ERK1/2 signaling pathway, whereas Rosamultin demonstrates anti-hypoxia effects via the PI3K/AKT signaling pathway. Additionally, Rosamultin plays an anti-radiation role through the Nrf2/HO-1 signaling pathway and exerts anti-inflammatory activity via the Wnt/β-catenin signaling pathway. Regarding Figure 13, it has been removed from the document.

  1. It would be useful to add a conclusion at the end of the manuscript.

Answer: A conclusion has been incorporated:In this study, the optimal extraction process for kajiichigoside F1 and rosamultin from the root of R. laevigata was established through a single-factor experiment combined with the Box-Behnken response surface methodology. Following this, an acute lung injury experiment was performed utilizing the extracted compounds, kajiichigoside F1 and rosamultin. The results demonstrated that both compounds have the potential to alleviate acute lung injury through the inhibition of the NF-κB signaling pathway. These findings establish a theoretical framework for further investigation into the molecular mechanisms underlying the anti-inflammatory properties of kajiichigoside F1 and rosamultin. Additionally, they provide a foundational basis for the prospective clinical application of these compounds in the treatment of acute lung injury.

  1. The manuscript is not well adapted to the journal. There is no pharmaceutical aspect. It would be more appropriate in Molecules (mdpi), for example.

Answer: On this topic, I will elucidate the following three aspects: (1) The root of the Cherokee rose is a traditional Chinese medicinal material and serves as a crucial component in Chinese patent medicines such as Jinji capsules and Sanjin tablets, highlighting its research significance. (2) Literature indicates that kajiichigoside F1 and rosamultin exhibit certain anti-inflammatory activities, as documented by Liang Y., Li L.Q., Wang L., Zhou L., and Yang X.S. in their study on the chemical constituents and anti-inflammatory activities of the rhizome of the ethnic medicine Rosa roxburghii (Guihaia, 2022, 42(09), 1531-1541). (3) In this study, an LPS-induced acute lung injury model was established, revealing that kajiichigoside F1 and rosamultin can mitigate inflammation in acute lung injury via the NF-κB signaling pathway.

  1. In the title, « Rosa laevigata » should be in italic and throughout the manuscript.

Answer: The term “Rosa laevigata” is now presented in italics.

  1. Abstract: “the content of kajiichigoside F1 and rosamultin and the total content of the two saponins” useless duplicated term.

Answer: The abstract has been revised to exclude the phrase: "the content of kajiichigoside F1 and rosamultin and the total content of the two saponins."

  1. Introduction: “Our research group began to study…” if a previous work has been done, indicate the reference.

Answer: Revisions have been made to address translation errors.

  1. Compounds should be defined properly and in a homogeneous form. kaji-ichigoside or kajiichigoside ? rosa-multinin (page 9) or rosamultin or rosamultinin ? The present mixture is not acceptable.

Answer: To ensure consistency in terminology, "kaji-ichigoside" has been standardized to "kajiichigoside F1," and "rosa-multinin" has been corrected to "rosamultin." Both kajiichigoside F1 and rosamultin are monomeric compounds.

  1. Provide complete analytical data for the two compounds in the Materials & Methods section (not in the main text).

Answer: Nuclear magnetic resonance data for kajiichigoside F1 and rosamultin are available.

Reviewer 3 Report

Comments and Suggestions for Authors

The manuscript explores the optimization of the preparation process of two triterpenoid saponins—kajiichigoside F1 and rosamultin—and investigates their protective effect against acute lung injury in an LPS-induced mouse model. The study holds merit because it focuses on natural compounds with potential therapeutic applications for inflammatory diseases. However, there are several flaws where the manuscript can be improved to enhance clarity, scientific rigour, and overall impact.

1. The abstract lacks details on the significance of the findings and their broader implications.

2. Some results are described without adequate statistical analysis. Particular emphasis is the NMR analyses. Compound 1 and compound 2 are isomers with a difference at C3. However, the reported chemical shifts are quite different. How can they be? Moreover, the authors measured the NMR spectra using different magnet NMR machines. For example, 400 MHz for compound 1 proton spectrum and for compound 2 C13- spectrum. Wny? Besides, only 1D-NMR may not be sufficient for the analyses of compound structure. The 2D-NMR spectral analysis is needed.

3. The discussion is somewhat repetitive and lacks depth in explaining the broader implications of the findings.

4. Many experimental details are not properly described in the manuscript. For example, in the single-factor experiment, the authors only wrote the conditions, but how they performed the purification is not seen. Similarly, the parameters for box-Behnken analysis using Design-expert software are not described.

5. The meaning and logic of Figure 13 is not clear 

Comments on the Quality of English Language

The English of the manuscript should be extensively revised.

Author Response

Thank you sincerely for your insightful comments and professional guidance. Your feedback has been instrumental in enhancing the academic rigor of our manuscript. In response to your suggestions and requests, we have made the necessary revisions to the manuscript. We are hopeful that these changes have further improved the quality of our work. Please find below our detailed response:

Response to Expert 3:

Thank you for your valuable suggestions. I will carefully consider your feedback to make the necessary revisions to the manuscript. Below are my responses to your comments:

  1. The abstract lacks details on the significance of the findings and their broader implications.

Answer: I have incorporated the summary content into the text.

  1. Some results are described without adequate statistical analysis. Particular emphasis is the NMR analyses. Compound 1 and compound 2 are isomers with a difference at C3. However, the reported chemical shifts are quite different. How can they be? Moreover, the authors measured the NMR spectra using different magnet NMR machines. For example, 400 MHz for compound 1 proton spectrum and for compound 2 C13- spectrum. Wny? Besides, only 1D-NMR may not be sufficient for the analyses of compound structure. The 2D-NMR spectral analysis is needed.

Answer: For personal reasons, I will elucidate the following: 400 MHz corresponds to the 1H nuclear magnetic resonance (NMR) spectrum, while 100 MHz pertains to the 13C NMR spectrum. The distinction between the two compounds, echinacoside and rosamultin, which are epimers differing at the 3-position with configurations of α-OH and β-OH, respectively, was determined by analyzing the coupling constants in the hydrogen spectrum and the chemical shift values in the carbon spectrum. Drawing upon previous literature, the coupling constants in the hydrogen spectrum and the chemical shift values in the carbon spectrum are explained as follows: In the ^1H-NMR spectrum, the primary differences are observed as follows: (1) When the 3-position is α-OH (echinacoside), the coupling constant J for H-3 ranges from 2.0 to 5.0 Hz; when the 3-position is β-OH, the coupling constant J for H-3 ranges from 6.0 to 10.0 Hz. (2) There is no significant difference in the chemical shift of the hydrogen spectrum between the two compounds.

  1. The discussion is somewhat repetitive and lacks depth in explaining the broader implications of the findings.

Answer: The discussion section has been restructured to first address the medicinal materials, extraction process, monomer properties, and pharmacological effects.

  1. Many experimental details are not properly described in the manuscript. For example, in the single-factor experiment, the authors only wrote the conditions, but how they performed the purification is not seen. Similarly, the parameters for box-Behnken analysis using Design-expert software are not described.

Answer: The single-factor experiment has been augmented:1. Extraction Method: Three samples of medicinal powder derived from R. laevigata, each with a precise weight of approximately 2 grams, were utilized. The extraction solvent was 60% ethanol, with a material-to-liquid ratio of 1:10, and the extraction was conducted over a period of 40 minutes. The extraction methods employed included ultrasonic extraction, heating reflux, and water decoction. Subsequently, the extraction yields of kajiichigoside F1 and rosamultin were quantitatively determined. Three parallel samples were prepared for each method, and the mean values were calculated. 2. Ethanol Volume Fraction: Three samples of R. laevigata medicinal powder, each weighing approximately 2 grams, were prepared with a material-to-liquid ratio of 1:10. Each sample underwent a single extraction process lasting 40 minutes. The ethanol volume fractions used were 20%, 40%, 60%, 80%, and 95%, respectively. The extraction yields of kajiichigoside F1 and rosamultin were subsequently measured. The procedure was performed in triplicate for each ethanol concentration, and the results were averaged. 3. Solid-Liquid Ratio: Three samples of R. laevigata medicinal powder, each weighing approximately 2 grams, underwent ultrasonic extraction with 80% ethanol. The extraction was performed for 40 minutes across different solid-liquid ratios of 1:10, 1:15, 1:20, 1:25, and 1:30 g/mL. The yields of kajiichigoside F1 and rosamultin were subsequently quantified. Each experimental condition was replicated three times, and the results were averaged to enhance accuracy and reliability. 4. Extraction Time: Three samples of R. laevigata medicinal powder, each precisely weighing approximately 2 grams, were subjected to ultrasonic extraction using 80% ethanol, maintaining a material-to-solvent ratio of 1:20 g/mL. The extraction was conducted for varying durations of 40, 60, 80, 100, and 120 minutes. Post-extraction, the yields of kajiichigoside F1 and rosamultin were quantified. Each experimental condition was replicated in triplicate, and the resulting data were averaged for subsequent analysis. 5.Extraction times: Three accurately weighed samples of R. laevigata medicinal powder, each approximately 2 grams, were employed in the study. The material-to-liquid ratio was consistently maintained at 1:20 g/mL, using 80% ethanol as the solvent for ultrasonic extraction, which was conducted over a period of 100 minutes. The extraction process was repeated one, two, and three times, respectively. Following the extraction, the yields of kajiichigoside F1 and rosamultin were quantitatively analyzed. Each experimental condition was performed in triplicate, and the results were averaged to ensure reliability.

The Box-Behnken design parameters included ethanol volume fraction, solid-liquid ratio, extraction duration, and frequency of extraction. The impact of these factors on the total concentrations of kajiichigoside F1 and rosamultin in R. laevigata was systematically investigated.

  1. The meaning and logic of Figure 13 is not clear 

Answer: Figure 13 has been removed.

Round 2

Reviewer 1 Report

Comments and Suggestions for Authors

Dear Authors

The only correction remained in title is "theirs" to "their." Otherwise, the MS is OK.

Dear Authors

The only correction remained in title is "theirs" to "their." Otherwise the MS is OK.

Reviewer 2 Report

Comments and Suggestions for Authors

My previous comments have been considered all and appropriate corrections have been made. The revised manuscript is significantly improved and clearer. I still think that it is not very appropriate for the journal, but OK for publication.

Comments on the Quality of English Language

ok

Reviewer 3 Report

Comments and Suggestions for Authors

The revised manuscript is greatly improved.